# Wnt-Dependent Activation of ERK Mediates Repression of Chondrocyte Fate during Calvarial Development

**DOI:** 10.3390/jdb9030023

**Published:** 2021-06-27

**Authors:** Beatriz A Ibarra, Cody Machen, Radhika P. Atit

**Affiliations:** 1Department of Biology, Case Western Reserve University, Cleveland, OH 44106, USA; bai9@case.edu (B.A.I.); czm8@case.edu (C.M.); 2Department of Genetics and Genome Sciences, Case Western Reserve University, Cleveland, OH 44106, USA; 3Department of Dermatology, Case Western Reserve University, Cleveland, OH 44106, USA

**Keywords:** craniofacial development, cell signaling, Wnt signaling, ERK signaling, cell fate

## Abstract

Wnt signaling regulates cell fate decisions in diverse contexts during development, and loss of Wnt signaling in the cranial mesenchyme results in a robust and binary cell fate switch from cranial bone to ectopic cartilage. The Extracellular signal-regulated protein kinase 1 and 2 (ERK1/2) and Wnt signaling pathways are activated during calvarial osteoblast cell fate selection. Here, we test the hypothesis that ERK signaling is a mediator of Wnt-dependent cell fate decisions in the cranial mesenchyme. First, we show that loss of *Erk1/*2 in the cranial mesenchyme results in a diminished domain of osteoblast marker expression and increased expression of cartilage fate markers and ectopic cartilage formation in the frontal bone primordia. Second, we show that mesenchyme Wnt/β-catenin signaling and *Wntless* are required for ERK activation in calvarial osteoblasts. Third, we demonstrate that Wnt and ERK signaling pathways function together to repress SOX9 expression in mouse cranial mesenchyme. Our results demonstrate an interaction between the Wnt and ERK signaling pathways in regulating lineage selection in a subset of calvarial cells and provide new insights into Wnt-dependent cell fate decisions.

## 1. Introduction

The development of the roof of the skull, the calvaria, involves several signaling pathways, differentiation program factors, and cellular movements in the supraorbital arch (SOA) mesenchyme (SOM) [1,2]. The frontal bone is derived from the cranial neural crest, whereas the parietal bone is derived from a combination of cranial neural crest and paraxial mesoderm progenitors [3]. Differentiation of intramembranous bone is unique from endochondral formation because the SOM condenses and differentiates directly into calvarial osteoblasts which give rise to the cranial bone/calvaria. Differentiation of the SOM and ossification of the calvaria begin basally and proceed apically, beginning with the anterior frontal bones then progressing to the posterior parietal bones during mammalian development [1,2]. The calvarial bone consists of two distinct domains, the outer periosteal layer and the inner endosteum [4]. Recent studies have identified the periosteum as a source of skeletal stem cells which are required for bone growth and repair [5]. It is unclear what signaling differences, if any, exist between the cells within the periosteum, endosteum, or other bone niches during early calvarial development.

Calvarial differentiation is regulated by a sequence of transcription factors and signaling pathways. The bone transcription program begins with expression of *Runx2* by calvarial osteoblast progenitors starting at E11.5 in the mouse. Ossification and differentiation of the bone progenitors into committed osteoblasts is specified by *Osx* downstream of Runx2 by E14.0. These osteoblasts secrete bone-specific extracellular matrix proteins which results in mineralized and ossified bone formation [3,6,7]. Signaling pathways, such as BMP, TGF-β, FGF, and Wnt and their target genes, have roles in endochondral and intramembranous bone differentiation and morphogenesis. Wnt signaling controls cell fate decisions in many contexts, including mammalian craniofacial development [1,2,8]. Wnt-dependent cell fate decisions are mediated through the canonical Wnt signaling pathway by transcriptional activity of β-catenin, or via the non-canonical Wnt/PCP or Wnt/Ca^2+^ signaling pathways. Importantly, canonical Wnt signaling is necessary in providing the osteo-inductive cue that initiates the calvarial differentiation program and inhibits cartilage fate [1,2,3]. Canonical Wnt signaling is initiated with the binding of a Wnt ligand to a cell-surface receptor, resulting in stabilization and translocation of β-catenin to the nucleus where it acts a transcriptional regulator with TCF/LEF transcription factors [8,9,10]. As mentioned earlier, loss of canonical Wnt signaling is characterized by a robust and binary cell fate switch from cranial bone to ectopic cartilage [11,12,13,14,15]. The role of noncanonical Wnt signaling also plays a role in osteogenic differentiation in vitro and in vivo [16,17,18]. Specifically, the noncanonical Wnt ligand, Wnt5a, is required for mechanical induction of osteogenic differentiation during wound repair [16], and activated Ror2 homodimerization resulting in increased osteoblast differentiation in human embryonic stem cells and murine calvarial explants in vitro [17]. While the role of Wnt signaling in calvarial bone development is well studied in both in vitro and in vivo, how Wnt signaling promotes differentiation of SOM to osteoblasts, while simultaneously repressing chondrocyte fate, is still unclear (reviewed in [3]). Identifying downstream regulators of Wnt signaling during osteoblast fate decisions will inform our understanding of how Wnt signaling regulates fate decisions in differentiation of other stem cell populations.

The ERK (extracellular signal-related kinase) signaling pathway cross-talks with Wnt/β-catenin signaling during endochondral osteoblast differentiation in vivo and in vitro, and may have roles in chondrogenic repression [19,20]. ERK1 and 2 are members of the MAPK (mitogen-activated protein kinase) signaling cascade and are typically activated by growth factor or G-protein ligand binding to a cell surface receptor [21,22,23]. Activation of ERK proteins, through phosphorylation by MEK (mitogen-activated protein kinase kinase, MAPKK), regulates several processes, such as cellular growth, proliferation, and differentiation [20]. The Wnt and ERK signaling pathways can act both up and downstream of each other in vitro and in vivo [19,20,24,25]. In NIH3T3 fibroblast cells, ERK signaling is activated downstream of Wnt signaling mediated by Wnt3a, independent of β-catenin, as well as by β-catenin/Tcf4 [24]. ERK signaling can in turn stabilize β-catenin by phosphorylating GSK3β, a key factor in the β-catenin degradation complex [20]. Matsushita et al. have reported diminished β-catenin levels following loss of ERK in limb-bud mesenchyme in vivo [19]. Whether ERK signaling is activated downstream of Wnt signaling in the cranial mesenchyme before cell fate decisions is not known.

Similar to Wnt signaling (reviewed in [4,26,27,28,29]), the ERK signaling pathway also has multiple roles during skeletal differentiation in vivo. Conditional loss of ERK activation in limb-bud mesenchyme results in loss of endochondral derived bone differentiation and ectopic cartilage in long bone perichondrium [19,23]. Similarly, *Prx1-cre;* β*-catenin^fl/fl^* embryos exhibit ectopic cartilage formation in the periosteum of long bones, ectopic cartilage and loss of bone mineralization in the skull and calvarial mesenchyme in vitro and in vivo [12]. These data suggest that β-catenin may repress cartilage differentiation through interaction with the ERK1/2 signaling pathway. The role of the ERK signaling pathway during chondrogenesis is unclear; however, the data suggest a temporally dynamic role based on several in vitro studies [20]. One study showed that ERK signaling is required for initial SOX9 activation in primary chondrocytes and C3H10T1/2 cells in vitro [30] while another noted that ERK activation during late chondrogenic differentiation resulted in increased chondrogenesis in limb bud mesenchyme, but only in the presence of p38/MapK signaling [31]. Recent studies have identified a unique relationship between ERK signaling and tracheal chondrogenic differentiation, demonstrating the existence of an incoherent feed forward loop where chondrogenesis is activated by SOX9 and simultaneously repressed by ERK [32]. The role ERK signaling plays during calvarial osteoblast fate decisions is unclear.

In this study, we test the hypothesis that ERK signaling is a downstream mediator of Wnt-dependent binary intramembranous bone-cartilage cell fate decisions. We found that ERK signaling is dispensable for cell-fate specification of calvarial osteoblasts, yet is required for calvarial bone formation by repression of chondrocyte fate in cranial bone progenitors. We use a suite of genetic tools to dissect the requirement of mesenchyme Wnt/β-catenin signaling and the other Wnt ligand pathways for ERK activation in the cranial mesenchyme during calvarial osteoblast fate decisions. These results demonstrate a genetic interaction between the Wnt and ERK signaling pathways to repress SOX9 and ERK signaling is activated downstream of Wnt signaling during calvarial osteoblast fate decisions in vivo.

## 2. Methods

### 2.1. Mice and Genotyping

Mouse lines used in the conditional functional studies include *En1Cre* (gift of Alex Joyner, [33]), *PDGFRαCreER* (Jax stock# 018280), *Wls flox* (gift of Richard Lang, [34]), β*-Catenin deleted* [35], β*-Catenin conditional floxed* [36]. *Erk1^−/−^* [37], and *Erk2^fl/fl^* [38] were graciously gifted from Dr. Guiming Liu. *PdgfrαCreER Erk1^−/−^ Erk2^fl/fl^* were maintained on a C57BL6 background, all other genetic combinations were maintained in a mixed genetic background. Mice were time-mated to yield desired crosses with vaginal plug day assigned as E0.5. Dams carrying embryos between E8.5 and E9.5 were orally gavaged with tamoxifen dissolved in corn oil (dosage specified with distinct crosses) to activate CreER and conditionally delete *Wls* or *Erk2* in the cranial mesenchyme. Embryos were harvested and processed for frozen sections as previously described [39]. For each experiment, a minimum of four mutants with litter-matched controls were studied unless otherwise noted. A minimum of three litters were used for each functional analysis. Case Western Reserve University Institutional Animal Care and Use Committee approved all animal procedures in accordance with AVMA guidelines (Protocol 2013–0156, Animal Welfare Assurance No. A3145–01).

### 2.2. Histology and Immunohistochemistry

Embryos were fixed in 4% paraformaldehyde at 4 °C, sucrose dehydrated, and embedded in O.C.T. Compound (Tissue-Tek Sakura, Sakura Finetek USA, Torrance, CA, USA) and sectioned at 14 μm as previously described [39,40]. H&E staining was performed by standard protocol.

Immunofluorescence (IF) on cryosections was performed as previously described [40], and sections were dried at room temperature, washed in 1X PBS or 1X PBS-T (PBS + 0.5% Tween), and blocked in goat serum or donkey serum. Primary antibodies were incubated overnight at 4 °C, washed the next day in 1X PBS or 1XPBS-T, incubated with species-specific secondary antibody (Alexa594, Invitrogen, Waltham, MA, USA) for one hour at room temperature, and then washed with DAPI (0.5 µg/mL), and mounted with Fluoroshield (Sigma F6057, St. Louis, MO, USA). The following primary antibodies were used: rabbit anti-Sox9 (1:1000, Chemicon AB5535, Japan), rabbit anti-Osx2 (1:2000, Abcam AB209484, Cambridge, United Kingdom), rabbit anti-Aggrecan (1:1000 for E13.5, 1:1500 for E16.5, Millipore AB1031, Burlington, MA, USA), rabbit anti-Lef1 (1:100, Cell Signaling Technology C12A5, Danvers, MA, USA), rabbit anti-Caspase3 (1:250, Abcam AB13847, Cambridge, United Kingdom), and rabbit anti-phospho-ERK1/2 (1:1250, Cell Signaling Technology 4370, Danvers, MA, USA). Wide-field images were captured at 20X and 40X using the Leica DMi8 inverted microscope using the Leica Application Suite X (LASX) software. Confocal images were captured on the Leica TCS SP8 gated STED 3X using a 40X oil immersion objective (HC PL APO 40X/1.30 NA CS2, Oil, FWD = 0.24 mm) detection by PMT detector or hybrid detectors and LASX software. Images were processed in Adobe Photoshop and Fiji/ImageJ [41,42].

For the p-ERK1/2 IF data, tyramide signal amplification using TSA plus Cyanine3.5 kit (PerkinElmer NEL763001KT, Akoya Biosciences, Marlborough, MA, USA) was used to amplify the signal. Cryosections were dried at room temperature, washed in PBS then underwent antigen retrieval at 60 °C for 20 min in a citrate buffer. Slides were washed in 1X TBS-T (TBS + 0.5%Tween-20) then blocked in 3% peroxide for 10 min, blocked with goat serum, and incubated in primary antibody overnight. The slides were then washed in 1X TBS-T and incubated in HRP-conjugated goat-anti-rabbit secondary (1:1500, PerkinElmer NEF812001EA, Akoya Biosciences, Marlborough, MA, USA) for 45 min. The slides were then incubated with the TSA Plus Working solution as per the manufacturer’s recommendations. Following amplification, the slides were washed with DAPI (0.5 µg/mL) and mounted with Fluoroshield (Sigma F6057, St. Louis, MO, USA). Images were captured on confocal microscopy as described above.

### 2.3. Immunofluorescence Quantification

Fiji/ImageJ was used to generate z-stacks for confocal images max-projected to the inner 5 microns. Each image was set to 8-bit and background removed with a 10-pixel “rolling ball”. The signal threshold was set to (13,255). “Watershed” was utilized to separate individual nuclei, which were then counted using the “Analyze Particles” function (circularity set to 0.15–1; particle size set to greater than 70 pixel units). For nuclear protein expression patterns (LEF1, SOX9, OSX), data were analyzed as a ratio of targeted nuclei to DAPI stained nuclei per region of the SOA (hand traced). For the non-nuclear signal (p-ERK1/2), total corrected fluorescence was measured using Fiji/ImageJ [43]. Total corrected fluorescence of non-nuclear signals was normalized to the number of DAPI-stained nuclei in the specified region. For all images, the boundary for the frontal bone primordia is defined on both the expression domain of OSX shown in adjacent sections (where applicable), as well as the morphological boundary between the condensed frontal bone mesenchyme as discerned by DAPI staining and the meningeal layers.

### 2.4. HCR–In Situ

HCR–in situ hybridization was performed on cryosections using HCR V3.0 bundle set designed by Molecular Instruments (https://www.molecularinstruments.com/products, Los Angeles, CA, USA) following the manufacturer’s protocol. The *Col2a1* probe utilized in the assay was designed by Molecular Instruments.

### 2.5. RT-qPCR

The relative expression of mRNA was analyzed using RT-qPCR as previously described [40]. Embryos at E12.5 or E13.5 were microdissected to isolate the cranial mesenchyme. The resulting tissue population is enriched cranial mesenchyme, following manual removal of the ectoderm and brain tissues. RNA was isolated as previously described [40,44]. Relative mRNA expression was quantified using 4 ng of cDNA on a StepOne Plus Real-Time PCR System (Life Technologies, Carlsbad, CA, USA) and the ΔΔCT method [45]. Commercially available TaqMan probes (Life Technologies, Carlsbad, CA, USA) specific to each gene were used: *Erk1* (Mm01278702_gH), *Erk2* (Mm01171077_m1), *Lef1* (Mm00550265_m1), *Tcf1* (mm00493445_m1), *Axin2* (Mm00443610_m1), *Sox9* (Mm00448840_m1), *OSX/Sp7* (Mm04209856_m1), *Runx2* (Mm00501584_m1), *Col2a1*, *Aggrecan* (Mm00545794_m1), β*-actin* (ActB, 4352933E), *Pdgfrα* (Mm00440701_m1), and *Wls* (Mm00509695_m1).

### 2.6. Digital Droplet PCR (DD-PCR)

DDPCR was used to quantify copy number of target transcripts. cDNA isolated as described above was used for all samples/targets. Reaction mixtures included ddPCR Supermix for Probes (no dUTP) (BioRad #1863023, Hercules, CA, USA), target probe (1 µL), and sample cDNA (4 ng/µL; 1 µL/reaction) to a final volume of 20 µL. Each reaction was loaded into a sample well of an eight-well disposable DG8 cartridge (BioRad # 1864008, Hercules, CA, USA) along with 70 µL Droplet Generation Oil for Probes (BioRad #1863005, Hercules, CA, USA) then covered with a gasket and loaded into a QX200 Droplet Generator (BioRad Laboratories, Hercules, CA, USA) to form droplets. Droplets were then transferred to a 96-well ddPCR plate (BioRad #12001925, Hercules, CA, USA), heat-sealed with foil, and amplified using a thermocycler using the following parameters: 95 °C for 10 min, 40 cycles of 94 °C for 30 s then 60 °C for 1 min, then 98 °C for 10 min. Completed PCR samples were loaded into the QX200 Droplet Reader (Bio-Rad Laboratories, Hercules, CA, USA) and droplets were analyzed per the manufacturer’s instructions. Data were analyzed using QuantaSoft software (Bio-Rad Laboratories, Hercules, CA, USA).

### 2.7. Protein Isolation and Immunoblotting

Enriched cranial mesenchyme was isolated as described above at E12.5 and protein was isolated as previously described [40]. Protein was isolated using RIPA buffer. Proteins were separated by SDS-PAGE using Mini-PROTEAN TGC gels (BioRad #456–1084, Hercules, CA, USA) or Mini-Protean TGX Stain-Free (BioRad #456-8084, Hercules, CA, USA). Western blots were performed with the following primary antibodies: rabbit-anti-p-ERK1/2 (1:500, Cell Signaling Technology 9101, Danvers, MA, USA), rabbit-anti-Erk1/2 (1:500, Cell Signaling Technology 9102, Danvers, MA, USA), and rabbit-anti-Lef1 (1:500, Cell Signaling Technology 2286, Danvers, MA, USA). Species-specific HRP-conjugated secondary antibodies were used at 1:10,000. Immunoblots were probed with anti-β-TUBULIN (1:400, Santa Cruz 9104, Dallas, TX, USA) as a loading control where indicated. Protein was detected using an Amersham ECL Western Blotting Analysis System (GE Healthcare RPN2109, Chicago, IL, USA), and developed onto film or imaged using a Biorad Chemidoc gel imager. Relative protein levels were quantified using Image J/Fiji to measure band intensity relative to the loading control (β-tubulin), or to total protein loaded per lane as measured by ImageLab analysis.

### 2.8. Statistics

All graphs and statistical analysis were generated using GraphPad Prism version 9.0.2 for Windows (GraphPad Software, San Diego, CA, USA). Data are presented as mean ± SEM in all graphs. All pairwise sample comparisons were performed using a Student’s t-test or Mann–Whitney test where indicated. The *p*-values for statistical tests in all figures are represented as * = *p* < 0.05, ** = *p* < 0.01, and *** = *p* < 0.001.

## 3. Results

### 3.1. ERK Signaling Is Required for Calvarial Bone Growth and Repression of Chondrocyte Fate In Vivo

To determine the role of ERK signaling in calvarial osteoblast fate decisions, we generated a cranial mesenchyme restricted loss-of-function ERK signaling mouse model. *PdgfrαCreER* is active broadly across the cranial mesenchyme as early as E9.0 as previously described [40,46,47]. We induced *CreER* recombination in the mouse cranial mesenchyme prior to calvarial osteoblast lineage decisions by administering tamoxifen at E8.5 and E9.5 (Figure 1A). This resulted in the efficient deletion of *Erk2* in an *Erk1* null background (CM-*Erk1/2*) (Figure 1B). We first queried cell fate in the cranial mesenchyme by analyzing expression of osteoblast and chondrocyte cell fate markers in the frontal bone primordia at E13.5 (Figure 1C,D). We found that loss of ERK1/2 did not alter the expression of the key osteoblast regulatory factor *Runx2* and *Osx* (*Sp7*) (Appendix A). Though ERK1/2 was not required for the differentiation of the frontal bone primordia into OSX^+^ osteoblasts (Figure 1E,F), we observed a decrease in the domain of OSX protein expression in the frontal bone primordia of CM-*Erk1/2* mutants compared to littermate controls. Next, we examined mRNA expression of cartilage determinants. Relative to controls, we found that *Sox9* mRNA was reduced by half and *Aggrecan* was elevated 3.5-fold in our CM-*Erk1/2* mutants relative to controls (Appendix A). Despite the significantly diminished *Sox9* gene expression, we observed ectopic expression of SOX9 and its downstream target, Aggrecan, within the basal region of the frontal bone primordia of the CM-*Erk1/2* mutants (Figure 1G,H). This ectopic expression of SOX9 and Aggrecan protein coincides with the formation of cartilage nodules within the frontal bone primordia of CM*-Erk1/2* mice, showing that ERK is required to repress chondrocyte differentiation within the cranial mesenchyme during osteoblast cell fate decisions. The discrepancy between mRNA and protein expression of key chondrocyte differentiation markers suggests a post-translational effect of ERK signaling on SOX9 protein stability or degradation [48].

We examined older embryos to determine if cartilage differentiation progresses in the CM-*Erk1/2* mice. At E16.5, the CM-*Erk1/2* mutant exhibits gross morphological defects including rounded skull, cleft lip/palate, stunted jaw and snout, polysyndactyly, and overall smaller stature (Appendix A). The combination of craniofacial defects observed here are similar to those observed in other congenital birth defects such as Robinow and Pierre Robin syndromes [49,50]. By H&E staining, the CM-*Erk1/2* mutant fail to form bone by histology in the frontal and parietal bone region, with complete loss of bone histology at the frontal bone region (Appendix A). Despite the lack of distinct cartilage cell morphology within the presumed frontal bone region, a subset of the CM-*Erk1/2* mutant mesenchyme in the medial periosteal region of the frontal bone continues to express ectopic SOX9 at E16.5, and SOX9 is expressed throughout the parietal bone (Appendix A). Together these data illustrate a specific requirement for ERK signaling in repressing SOX9 and chondrocyte fate during calvarial development.

Due to the severity of the phenotype in CM-*Erk1/2* mutants, we observed nearly 75% reabsorption and loss of pregnancy. To increase the success rate of pregnancies and viability of mutant embryos, we lowered the tamoxifen dosage, which resulted in a 60–70% reduction in *Erk2* expression (CM-*Erk1/2*^KD^) (Figure 2A,B). In contrast to CM-*Erk1/2* mutants, the domain of OSX protein expression was comparable between controls and CM-*Erk1/2*^KD^ (Figure 2E,F), demonstrating that 70% *Erk* expression is sufficient to form frontal bone. Notably, there is a twofold increase in the number of SOX9+ cells within the frontal bone primordia of CM-*Erk1/2*^KD^ embryos relative to littermate controls (Figure 2G,H,M). Ectopic expression of SOX9 was consistent with the misexpression of the downstream cartilage marker, *Col2a1,* in the frontal bone primordia (Figure 2I,J). Expression of the late chondrocyte marker, Aggrecan, is comparable to controls (Figure 2K,L). We did not observe the formation of cartilage nodules in the CM-*Erk1/2*^KD^ embryos, suggesting that while ERK1/2 is required to repress SOX9 protein expression, the conversion to differentiated chondrocyte fate may be dependent on the level of ERK present in the cell. Together, these data suggest that *Erk2* activation is required in a dose-dependent manner to regulate calvarial growth and is required for repression of cartilage determinants during early calvarial osteoblast differentiation, by a mechanism to be determined in future studies.

### 3.2. Canonical Wnt Signaling Is Required for Activation of ERK Signaling In Vivo

Next, we investigated whether ERK signaling is activated downstream of Wnt/β-catenin signaling during calvarial osteoblast cell fate decisions in vivo. We utilized a mouse model in which β-catenin is conditionally deleted in the cranial mesenchyme beginning at E9.5 using *En1Cre* as previously described [13]. Without β-catenin, the mesenchyme cannot transduce canonical Wnt signaling (Figure 3A) and the cranial mesenchyme cells undergo a cell fate switch from osteoblast to chondrocyte [13]. Compared to E12.5 *En1Cre;* β*-catenin^Δ/+^* control, phosphorylation of ERK1/2 proteins was diminished by 92% in *En1Cre;* β*-catenin^Δ/fl^* (CM-β-cat^Δ/fl^) cranial mesenchyme (Figure 3B–D). Total ERK1/2 protein levels remained unchanged, indicating that the decrease in pERK1/2 was due to diminished activation of the ERK signaling pathway rather than changes in transcription of *Erk1/2* genes by β-catenin. Spatial analysis of ERK1/2 phosphorylation by immunofluorescence confirmed that the pERK1/2 is broadly expressed across the cranial mesenchyme in control embryos, and is diminished in CM-β-cat^Δ/fl^ (Figure 3F–I’). These data demonstrate that β-catenin is required for activation of ERK1/2 in the frontal bone primordia during the window of osteoblast cell fate specification in vivo.

β-catenin is primarily recognized for its importance in transducing canonical Wnt signaling, but also functions independent of Wnt signaling during cellular adhesion and cadherin signaling [51]. To confirm that the observed decrease in pERK1/2 is due to loss of canonical Wnt signaling, and not an alternative function of β-catenin, we utilized an alternative conditional loss-of-function model of Wnt signaling in the cranial mesenchyme. *WNTLESS* (*Wls*) is a transmembrane protein that is specifically required for secretion of all Wnt ligands [34,52,53]. Using *Crect*, a surface ectoderm restricted Cre line [15,54], we conditionally deleted *Wls* in the surface ectoderm which results in complete loss of Wnt signaling in the underlying cranial mesenchyme while leaving β-catenin intact (Appendix A) [15]. In this model, we also observe complete loss of pERK expression in the mutant cranial mesenchyme (Appendix A). These data demonstrate that canonical Wnt/β-catenin signaling is required for ERK activation in the cranial mesenchyme during osteoblast fate decisions.

### 3.3. Mesenchyme Wnts Are Required for ERK1/2 Activation in the Cranial Mesenchyme

We next investigated the role of mesenchyme Wnt ligands on ERK1/2 phosphorylation. Notably, Wnt5a and Wnt11, the classical non-canonical Wnt ligands, are broadly expressed in the cranial mesenchyme, making this model a useful tool to study the contribution of non-canonical Wnt signaling [15]. Therefore, we conditionally deleted *Wls* in the cranial mesenchyme using *PdgfrαCreER* as previously described [34,55] (CM-*Wls*) (Figure 4A,B). At E12.5 protein expression of Lef1, a readout of canonical Wnt signaling, was comparable in *Cre-; Wls^fl/fl^* controls and CM-*Wls*, indicating that canonical Wnt signaling remained intact at E12.5 (Figure 4C–E). Next, we determined if phosphorylation of ERK1/2 is diminished in the cranial mesenchyme of the CM-*Wls*. We observed a 90% reduction in pERK1/2 levels in the cranial mesenchyme of CM-*Wls* embryos compared to controls (Figure 4C–E). Total ERK1/2 protein expression levels were comparable in controls and CM-*Wls*. These data show that mesenchyme Wnt ligand secretion is required to activate ERK signaling in the cranial mesenchyme.

We next explored whether CM-*Wls* resulted in discriminate changes in the level of canonical Wnt signaling before cell fate decisions in the cranial mesenchyme. Wnt ligands, inhibitors, and target genes are patterned in distinct domains in the cranial mesenchyme. Graded expression of nuclear β-catenin protein, *Axin2* mRNA, and *Lef1* mRNA in the cranial mesenchyme between E11.5–12.5 suggest that ectoderm Wnts contribute to a gradient of Wnt signaling response in the mesenchyme [15]. In addition, Wnt responsive target genes across tissues, such as *Tcf1* and *Lef1*, require different levels of Wnt signaling for transcription [56,57]. To query for subtle and dynamic changes in levels of Wnt signaling in the cranial mesenchyme of the CM-Wls mutant, we used droplet digital PCR (DDPCR) to measure discrete copy numbers of *Axin2*, *Tcf1*, and *Lef1* mRNA. We saw a significant decrease in copy number of *Tcf1* (−33%) and *Axin2* (−25%) in our mutants compared to the controls, while there was no effect in *Lef1* copy number (Appendix A). The significant changes in *Axin2* and *Tcf1* expression suggest modest and detectable change in the levels of Wnt activation in the cranial mesenchyme at E12.5. All together, we show that loss of mesenchyme *Wls* results in a disruption of Wnt signaling response in the mesenchyme visualized by the significant changes in Wnt target gene expression. These changes are sufficient to diminish activation of ERK through an unknown mechanism.

As development proceeds, the calvarial bone primordia develops away from the ectoderm and needs to generate its own Wnt ligands for canonical Wnt signaling and non-canonical Wnt ligand expression as well as osteoblast differentiation [15]. At E13.5, we begin to see diminished expression of Lef1 within the frontal bone primordia of CM-*Wls* embryos relative to controls (Figure 4G,H). In the controls, SOX9 expression is restricted to the cartilage base and faintly in the basal meninges (Figure 4I). In CM-Wls, SOX9 is ectopically expressed in the medial periosteum of the basal frontal bone primordia accompanied by ectopic cartilage formation and diminished frontal bone formation by E14.5 (Figure 4I–L). Furthermore, pERK1/2 is significantly decreased in the cranial mesenchyme of CM-Wls embryos relative to controls (Figure 4M–O). These data demonstrate a conserved yet delayed cell fate switch phenotype as observed within the CM-β-cat^Δ/fl^ mutant (Figure 4P). Together, these results suggest a requirement of different arms of the Wnt signaling for activation of ERK1/2 in the cranial mesenchyme during calvarial development and repression of SOX9 and cartilage fate (Figure 4P).

### 3.4. β-Catenin and ERK1/2 Signaling Can Interact to Repress SOX9 Expression in Cranial Mesenchyme

Finally, to test if β-catenin and the ERK signaling pathway genetically interact in the cranial mesenchyme, we generated a novel compound heterozygous conditional mouse model for the β*-catenin*, *Erk1* and *Erk2* alleles (Figure 5A) using both *PDGFRαCreER* and *En1Cre* lines. Individual heterozygous mutants for either *Erk1*, *Erk2*, or β*-catenin* proceed with normal calvarial development (Appendix A, Figure 3A,F) [13,58]. Relative to the *Cre*- controls, we found ectopic expression of SOX9 protein in the most medial region of the frontal bone primordia (OSX + cells) in two different *Cre*+ triple compound heterozygotes (Figure 5B–H’’’). The spatial localization of ectopic SOX9 resembles the CM-*Erk1/2*^KD^ and CM-*Wls* mutants, indicating that heterozygosity of the β*-catenin* and *Erk1/2* alleles is sufficient to phenocopy loss of ERK and loss of mesenchyme Wnts at the level of SOX9 repression (Figure 2G,H and Figure 4I,J). These results indicate that both Wnt/β-catenin and ERK signaling pathways are simultaneously required to repress SOX9 during calvarial bone development in mice. Our data show that ERK signaling is highly responsive to variations in cranial mesenchyme Wnt signaling, and propose a model wherein ERK is activated downstream of Wnt signaling and mesenchyme Wnts to repress SOX9 during calvarial cell fate specification in mouse (Figure 4P).

## 4. Discussion

Wnt-dependent repression of cartilage fate during calvarial development is poorly understood. Using a suite of conditional genetic mouse models, our data suggest a new working model that identifies ERK signaling as a key downstream mediator for Wnt-dependent chondrocyte repression in a subset of calvarial osteoblast precursors during cell fate selection. ERK signaling activation is dependent on β-catenin and Wnt ligands in the cranial mesenchyme. Our in vivo genetic mutants also demonstrate that β-catenin and ERK1/2 interact genetically to repress SOX9 expression and chondrocyte fate in the cranial mesenchyme in vivo.

Our in vivo analysis shows that Wnt signaling is required for ERK pathway activation in the cranial mesenchyme, but how is unclear. β-catenin is a transcriptional regulator which binds to TCF/LEF to initiate transcription during canonical Wnt signaling. Downstream of canonical Wnt signaling, and a putative candidate for Wnt/β-catenin ERK activation, are the *Twist* genes. Our lab has demonstrated that expression of the transcription factors *Twist1/Twist2* is regulated by Wnt/β-catenin signaling. Although Twist1 represses *Sox9* mRNA expression [14,59], conditional deletion of *Twist1* in the cranial mesenchyme only results in a partial phenocopy of cartilage conversion of the occipital bone [14]. Twist can activate ERK signaling in vitro and overexpression of Twist is sufficient to increase ERK activation during EMT in breast cancer cells [60]. However, these data do not demonstrate that Twist1 is necessary for ERK pathway activation. Future studies will be necessary to understand the mechanism of Wnt/β-catenin ERK activation and to determine if Twist1 is required for ERK activation during calvarial osteoblast fate decisions in vivo.

Our results suggest that mesenchyme Wnts mediating β-catenin-independent Wnt signaling may function in either a parallel or linear pathway to canonical Wnt/β-catenin-dependent signaling to activate ERK signaling (Figure 5C–E and Appendix A). Using CM-*Wls* mutants, we were able to further dissect the role of Wnt signaling beyond canonical Wnt/β-catenin signaling. In support of a linear pathway, we previously showed that canonical Wnt signaling is required to transcriptionally regulate non-canonical Wnt ligands Wnt5a and Wnt11 in the cranial mesenchyme, suggesting that non-canonical Wnt signaling is downstream of canonical Wnt signaling [15]. Our results with *PDGFRαCreER* deletion of *Wntless* support a linear pathway because there is a temporal delay in the observed cell fate switch to chondrocyte fate in the CM-*Wls* mutants (Figure 5). Alternatively, our data also provide some evidence towards a parallel pathway. At E12.5, canonical Wnt signaling is comparable to controls in the CM-Wls mutants by LEF1 protein expression. Yet, we observe significantly decreased levels of ERK activation in our CM-*Wls* embryos (Figure 5). These data demonstrate a requirement of mesenchyme Wnts, independent of canonical Wnt signaling, for ERK activation during early calvarial cell fate decisions. In this model of parallel pathways, both mesenchyme Wnts and β-catenin are required to activate ERK signaling and converge at the point of SOX9 expression, but whether they negatively regulate SOX9 expression by the same mechanism is unknown. Another consideration is that the canonical and non-canonical Wnt signaling pathways may exist within a feed-forward loop to activate ERK signaling. Our CM-*Wls* embryos show significant but modest decreases in two canonical Wnt signaling target genes, *Axin2* and *Tcf1*. These decreases may indicate that canonical Wnt signaling is just beginning to decrease by E12.5 as a result of spatial differences as the mesenchyme grows away from the ectoderm, or as a result of changes in a gradient of Wnt signaling present in the cranial mesenchyme following loss of mesenchyme Wnts. Further studies will be necessary to understand how mesenchyme Wnt ligands and non-canonical Wnt signaling change throughout the cranial mesenchyme to activate the ERK signaling pathway, and how they intersect with canonical Wnt/β-catenin dependent activation of ERK. Future studies using specific mouse models targeting the functions of the Wnt/PCP and Wnt/Ca^2+^ signaling are necessary to test the roles of the non-canonical Wnt signaling pathway on ERK activation in the cranial mesenchyme in vivo.

The role of Wnt signaling on the ERK signaling pathway is well studied in individual cell types in vitro. However, there are several limitations in understanding how these two signaling pathways guide calvarial osteoblast fate decisions from in vitro studies which do not capture the complexity of the environment in which the calvaria develops in vivo. Unlike long bones, calvarial osteoblasts develop between the brain and meningeal mesenchyme medially, and dermal fibroblasts and surface ectoderm laterally. The close proximity of each cell type results in cross-talk of cell signaling and communication between cell populations. Furthermore, there are mechanical cues and regional differences in the composition and signaling of the extracellular matrix in the cranial mesenchyme as the cranial bones expand apically [61]. These regional differences can impact how cell fate decisions are made and are difficult to recapitulate in vitro, leading to an incomplete understanding of cell signaling mechanisms across tissue types. Our in vivo studies demonstrate that ERK signaling is required to repress the chondrocyte determinants in a subset of calvarial osteoblasts in a dose-dependent manner. The spatial restriction of this phenotype is limited to a small subset of cells located in the medio-basal region of the frontal bone primordia and corresponds to the future periosteum of the frontal bone, suggesting heterogeneity among the cells. It is not clear what differences exist within the signaling requirements for this subset of cells. This effect is not restricted to the frontal bone; however, the level of ectopic SOX9 observed in the E16.5 CM-*Erk1/2* mutant is more subtle, suggesting that there may be variability in the survival of the calvarial osteoblasts. Our data suggest a conserved role in the function of mesenchyme Wnts and ERK signaling on cell fate regulation within this subpopulation of calvarial osteoblasts. The phenotype of the CM-*Wls* mutants resembles CM-*Erk* mutants in the localization of ectopic SOX9 and cartilage nodule formation. Spatial segregation of Wnt ligands, Wnt inhibitors combined with graded expression of Wnt signaling target genes, suggests there can be regional variation in the level of Wnt signaling experienced by individual calvarial osteoblasts [15,56]. The subset of cells within the region where ectopic SOX9 is observed in our CM-*Erk* and CM-*Wls* mutants may experience less canonical Wnt signaling due to the high levels of the Wnt inhibitor DKK2, and the increased distance from the ectoderm Wnts [15]. Regional variation in levels of Wnt signaling may contribute to the heterogeneity of the calvarial osteoblasts and their ability to repress the cartilage fate. Currently, quantification of Wnt signaling at the cellular level is not possible in vivo. Techniques such as single-cell sequencing can be utilized in the future to reveal if there are differences amongst the calvarial osteoblasts that contribute to the periosteum, endosteum, and osteogenic front and identify the differences in Wnt and ERK signaling levels.

In summary, our data reveal that Wnt/β-catenin signaling can activate ERK signaling to ensure a subset of calvarial osteoblasts are specified towards the intramembranous differentiation and repress the cartilage fate. We show a distinct requirement of ERK signaling in cell fate specification during early calvarial development, and demonstrate cross-talk between Wnt signaling and ERK signaling on cartilage repression that is required for normal calvarial development. Our results provide evidence towards identifying a specialized subset of periosteal cells that are highly sensitive to signaling dynamics of the Wnt and ERK signaling pathways. Our conclusions provide broad insights into Wnt-dependent cell fate specification that will be informative for intramembranous bone repair and regeneration.

## Figures and Tables

**Figure 1 jdb-09-00023-f001:**
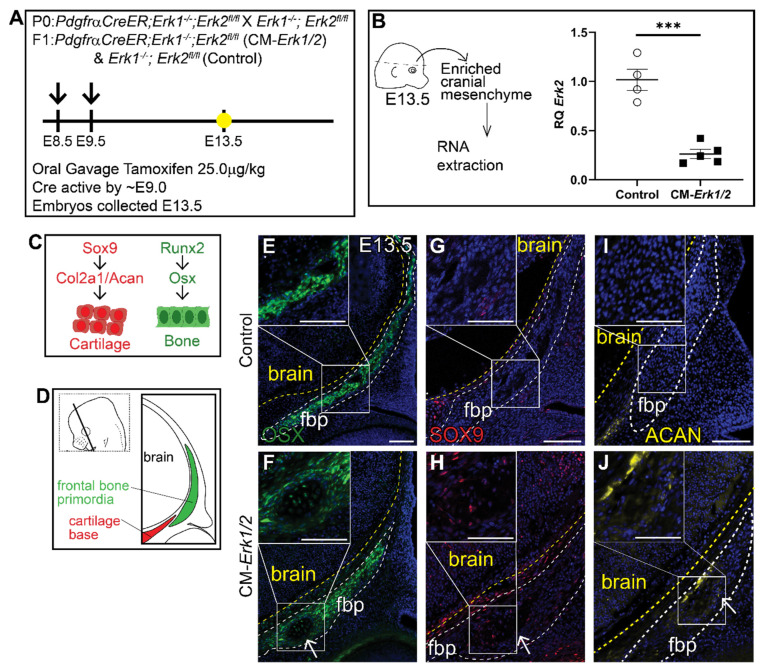
**Loss of ERK1/2 results in diminished expression of OSX, and ectopic expression of chondrocyte fate markers at E13.5.** Schematic depicting the gavage regimen and timing of deletion of *Erk2* (**A**). (**B**) Schematic depicting the work flow and tissue source for the RNA analysis, RT-qPCR analysis depicting the relative quantity of mRNA transcripts for *Erk2* demonstrating deletion efficiency of gavage regimen. *p*(***) = 0.0002. Bone and cartilage cell fate and marker schematic (**C**) and orientation and plane of specimens (**D**). Immunofluorescence for calvarial progenitor marker, OSX (**E**,**F**), and chondrocyte markers, SOX9 (**G**,**H**) and Aggrecan (ACAN) (**I**,**J**) at E13.5. Arrows indicate ectopic cartilage formation. fbp, frontal bone primordia; *n* = 3–5.

**Figure 2 jdb-09-00023-f002:**
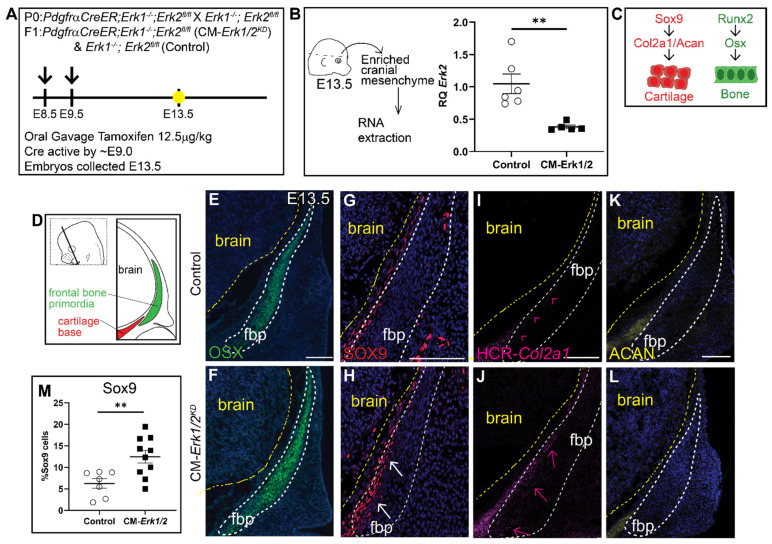
**Knockdown of *Erk1/2* results in ectopic expression of SOX9 and *Col2a1* but not OSX or Aggrecan.** (**A**) Schematic depicting the gavage regimen and timing of deletion of *Erk2*. (**B**) Schematic depicting the work flow and tissue source for the RNA analysis, RT-qPCR analysis depicting the relative quantity of mRNA transcripts for *Erk2* demonstrating deletion efficiency of gavage regimen. 70% knockdown of *Erk2* is achieved with a minimal gavage dosage (1.25 mg/kg Tamoxifen). *p*(**) = 0.0036 (**C**) Bone and cartilage cell fate and marker schematic. (**D**) Schematic indicating the orientation and plane of specimens. (**E**,**F**) Immunofluorescence of OSX at E13.5. (**G**,**H**) There is increased SOX9 expression in the frontal bone primordia of *Erk1/2* mutants relative to controls. (**I**,**J**) HCR-*Col2a1* in situ stain in CM-*Erk1/2* mutants and controls. (**K**,**L**) Aggrecan (ACAN) immunofluorescence shows no change in the downstream marker of SOX9 at E13.5. (**M**) Percent SOX9+ cells is increased in the mutant relative to the control. White arrows indicate presence of ectopic SOX9 within the fbp. Pink arrowheads and arrows point to edge of *Col2a1* expression boundary in the control and mutant, respectively. fbp, frontal bone primordia; *n* = 3–10.

**Figure 3 jdb-09-00023-f003:**
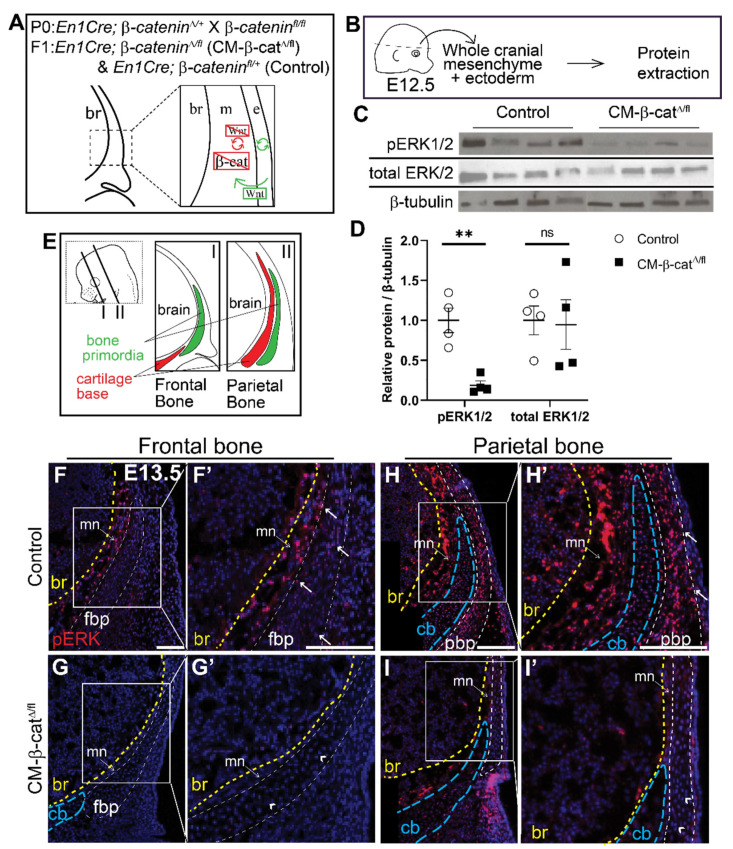
**ERK signaling is diminished in CM-**β**-cat**^Δ**/fl**^**mutants.** (**A**) Schematic depicting the status of Wnt signaling in the cranial mesenchyme and overlying ectoderm in this mouse model of Wnt signaling deletion. br, brain; m, mesenchyme; e, ectoderm. β-catenin is deleted in the cranial mesenchyme preventing mesenchyme Wnt signaling. The ectoderm is able to secrete Wnt ligands and undergo Wnt signaling. (**B**) Schematic depicting the work flow and tissue source for the protein analysis in panels C and D. (**C**,**D**) Expression of activated ERK1/2 (pERK1/2) and total ERK1/2 relative to β-tubulin expression in *En1Cre;*β*-catenin^fl/+^* (control) and *En1Cre;*β*-catenin*^Δ*/fl*^ (CM-β-cat^Δ/fl^) normalized to control. *p*(**) = 0.0026 (**E**) Schematic indicating the plane and orientation of specimens. (**F**–**I’**) Immunofluorescence staining of pERK1/2 in control and CM-β-cat^Δ/fl^ coronal sections at E13.5 in the frontal (**F**–**G’**) and parietal bone regions (**H**–**I’**). Yellow lines demarcate the br, brain; blue lines demarcate the cb, cartilage base; mn, meninges. Arrows denote pERK1/2 signal in the bone primordia of the controls and arrowheads denote lack of signal in the CM-β-cat^Δ/fl^. fpb, frontal bone primordia; pbp, parietal bone primordia; *n* = 4.

**Figure 4 jdb-09-00023-f004:**
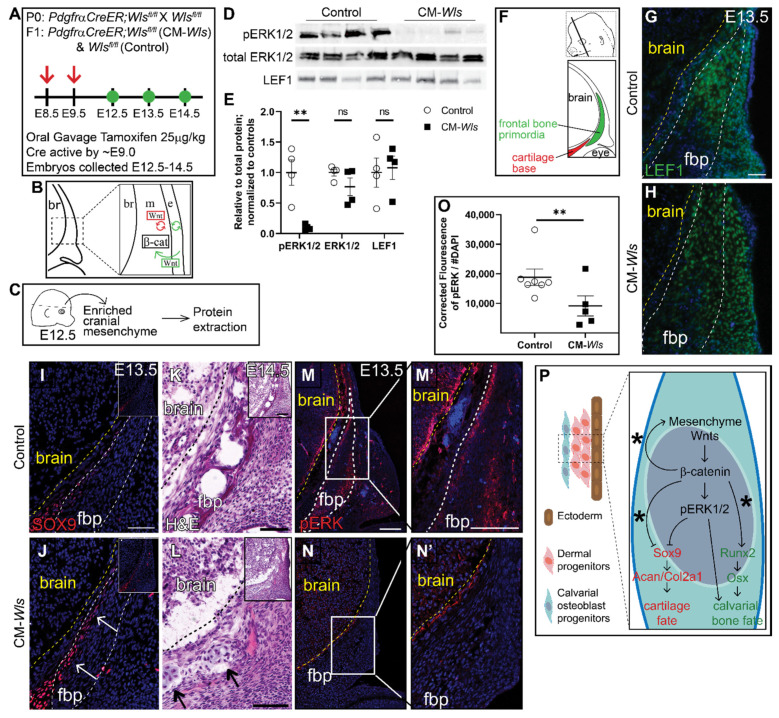
**Mesenchyme Wnts are required for ERK1/2 activation in vivo.** (**A**) Schematic depicting the gavage regimen. (**B**) Schematic depicting the status of Wnt signaling in the cranial mesenchyme and overlying ectoderm in this mouse model of Wnt signaling deletion. br, brain; m, mesenchyme; e, ectoderm. β-catenin remains intact within the cranial mesenchyme. The ectoderm is able to secrete Wnt ligands and undergo Wnt signaling. The mesenchyme cannot secrete Wnt ligands but can transduce Wnt signaling from the Wnt ligands secreted by the overlying ectoderm. (**C**) Schematic depicting the work flow and tissue source for the protein analysis. (**D**,**E**) Expression of activated ERK1/2 (pERK1/2), total ERK1/2, and LEF1 relative to total protein and normalized to controls in E12.5 control and CM-*Wls*. *n* = 4 biological replicates. *p*(**) = 0.0026. (**F**) Schematic indicating the plane and orientation of specimens. (**G**,**H**) Expression of LEF1 is decreased fbp of *Pdgfr*α*CreER;Wls^fl/fl^ (*CM-*Wls)* relative to controls. (**I**,**J**) CM-*Wls* express ectopic SOX9 in the fbp relative to controls. (**K**,**L**) H&E shows ectopic cartilage nodules in CM-*Wls* but not controls. (**M**–**N’**) Immunofluorescence of control and CM-*Wls* showing diminished pERK1/2 in the CM-*Wls* relative to the controls. (**O**) Quantification of pERK1/2 IF by corrected fluorescence using ImageJ. *n* = 7 controls; 5 mutants. *p*(**) ≤ 0.01. (**P**) Summary schematic illustrating the role of Wnts on ERK activation and downstream effect on cell fate decisions in calvarial osteoblast progenitors. (*****): Goodnough et al 2016. Arrows point to ectopic cartilage nodule. Yellow line demarcates brain in IF panels; black line demarcates brain in panels K & L. fbp, frontal bone primordia.

**Figure 5 jdb-09-00023-f005:**
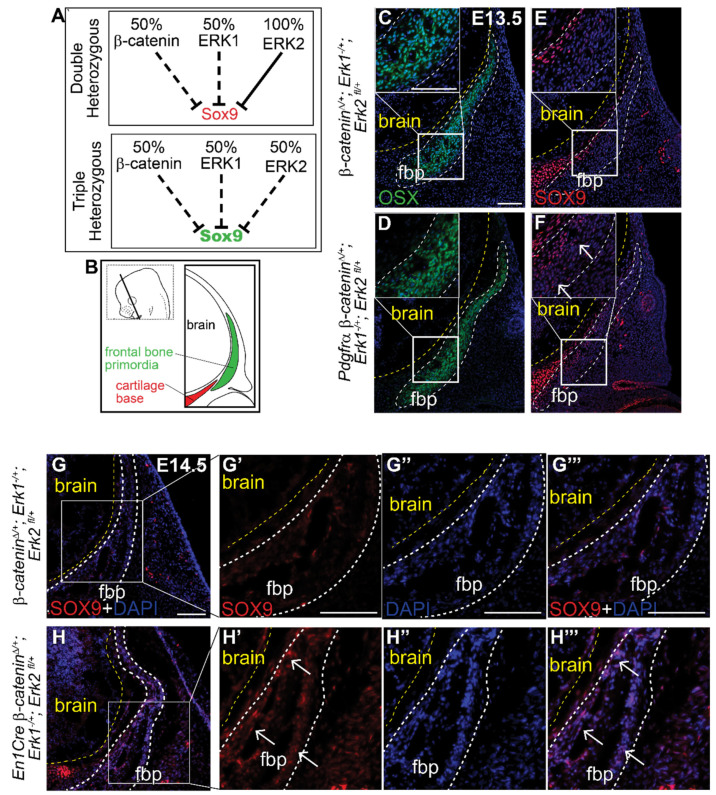
**Triple heterozygous β-*catenin*;*Erk1/2* mutants demonstrate genetic interaction between the Wnt and ERK signaling pathways in SOX9 transcription.** (**A**) Schematic describing genetic interaction hypothesis between Wnt and ERK signaling pathways to repress SOX9. (**B**) Schematic illustrating coronal sections. Immunofluorescence of OSX (**C**,**D**) and SOX9 (**E**,**F**) at E13.5 in β*-catenin*^Δ*/+*^*; Erk1^−/+^; Erk2 ^fl/+^* and *Pdgfr*α β*-catenin*^Δ*/+*^*; Erk1^−/+^; Erk2 ^fl/+^* heterozygous embryos. Immunofluorescence of SOX9 at E14.5 in β*-catenin*^Δ*/+*^*; Erk1^−/+^; Erk2 ^fl/+^* and *En1Cre* β*-catenin*^Δ*/+*^*; Erk1^−/+^; Erk2 ^fl/+^* embryos (G–H’’’). Yellow line demarcates brain, white dashed lines demarcate fbp and region of interest. Arrows identify ectopic SOX9 expression within the fbp. fbp, frontal bone primordia; *n* = 3.

## Data Availability

Data is contained within this article and the Appendix A.

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
