# Peer review of "Wnt-Dependent Activation of ERK Mediates Repression of Chondrocyte Fate during Calvarial Development"

_jdb, 2021, doi:10.3390/jdb9030023_

Round 1

Reviewer 1 Report

The study represents attempts by the authors to better understand development of the calvarial bones by dissecting and tying together three major signaling pathways – bone transcriptional, Wnt (canonical and non-canonical), and ERK signaling pathways during osteoblast differentiation and chondrocyte repression in the development of the calvarial bones. The authors’ hypothesis was that ERK signaling is a downstream mediator of Wnt-dependent binary intramembranous bone-cartilage cell fate decisions. They found that ERK signaling is dispensable for cell-fate specification of calvarial osteoblasts; however, this signaling is required for calvarial bone formation by repression of chondrocyte fate in cranial bone progenitors. Using various mouse mutants, the authors showed a genetic interaction between the Wnt and ERK signaling pathways to repress Sox9 and ERK signaling is activated downstream of Wnt signaling during calvarial osteoblast fate decisions in vivo.

As a whole, the manuscript is well written, with clear figures and explanations. Their work is a follow-up from the lab in the area of cranial bone development. The data presented in the manuscript are of high quality overall. There are some inconsistencies and lack of details in some areas.

Comments:

  1. In some experiments, data of the frontal bone were reported whereas in some, the parietal bone was included. The data of the parietal bone were not as clear. Does the fact that the frontal and parietal bones are of different origins (neural crest versus combination of neural crest + non-neural crest) have an impact in the signaling mechanisms being studied?
  2. There appeared to be a dose response of the levels of ERK on the expression of Sox9 and aggrecan, as shown by the loss-of-function ERK signaling versus the 60-70% reduction of ERK1/2 mutants. For example, there was decreased Sox9 expression (but increased Aggrecan protein), but a 2-fold increase in number Sox9+ cells (and no change in Aggrecan protein) in the full and 60-70% reduction ERK1/2 mutants. Please clarify.
  3. The E16.5 CM-ERK1/2 mutant embryos reportedly had rounded skull, facial cleftings, jaw malformation, limb defects and smaller stature – what are the implications of these phenotypic changes? Likewise, statements such as “lacked mature bone by histology” lacked details both in the presentation of the data and description. Fig. 2 – difficult to see the details, e.g., D, E, F, G, K – resolution not high enough. Any PCR to see the change in expression? Parietal bone hard to detect changes. Although overall the images are of high quality and presentation; however, some of the images, especially of the parietal bone, are difficult to discern.
  4. The schematic presented on Fig 5 P appears rather simplistic. Can more of the data generated in the study be incorporated in it?

Author Response

Point 1: In some experiments, data of the frontal bone were reported whereas in some, the parietal bone was included. The data of the parietal bone were not as clear. Does the fact that the frontal and parietal bones are of different origins (neural crest versus combination of neural crest + non-neural crest) have an impact in the signaling mechanisms being studied?

Response to Point1: Our data are consistent with previous studies showing the signaling requirements differ between the frontal and parietal bones. We have addressed this concern in the discussion. (Page 16, line 18)

Point 2: There appeared to be a dose response of the levels of ERK on the expression of Sox9 and aggrecan, as shown by the loss-of-function ERK signaling versus the 60-70% reduction of ERK1/2 mutants. For example, there was decreased Sox9 expression (but increased Aggrecan protein), but a 2-fold increase in number Sox9+ cells (and no change in Aggrecan protein) in the full and 60-70% reduction ERK1/2 mutants. Please clarify.

Response to Point 2: We absolutely agree with the observation of a dose-dependent effect of ERK1/2, and have revised the text and nomenclature to address this effect (Page 7 line 7, Page 8 line 2-6).  We observe that ERK signaling is required in a dose-dependent manner for calvarial osteoblast differentiation, but not specification.  Furthermore, we observe ectopic cartilage formation in the CM-Erk1/2 mutant embryos, but not in the CM-Erk1/2KD, despite the presence of ectopic Sox9 in both genotypes. We believe that this may be due to the heterogeneity of osteoblasts with varying levels of osteo-chondrogenic determinants to translate to a phenotypic conversion of osteoblast to chondrocyte. Future studies could use single-cell RNAseq to determine discrete changes in the expression profiles of the frontal bone osteoblasts to confirm this hypothesis.

Point 3: The E16.5 CM-ERK1/2 mutant embryos reportedly had rounded skull, facial cleftings, jaw malformation, limb defects and smaller stature – what are the implications of these phenotypic changes? Likewise, statements such as “lacked mature bone by histology” lacked details both in the presentation of the data and description. Fig. 2 – difficult to see the details, e.g., D, E, F, G, K – resolution not high enough. Any PCR to see the change in expression? Parietal bone hard to detect changes. Although overall the images are of high quality and presentation; however, some of the images, especially of the parietal bone, are difficult to discern.

Response to Point 3: We have added in the clinical implications towards the CM-Erk1/2 phenotype and have clarified the descriptions as requested (Page 7 line 7, 12).  We cannot generate additional E16.5 stage embryos due to high rates of miscarriage after E13.5 in dams carrying PdgfraCreER; Erk1-/-; Erk2fl/fl embryos. We have taken higher resolution photos to replace the H&Es in Figure2 Panels D-G, extensively revised the organization and moved this figure to Supplement. 

Point 4: The schematic presented on Fig 5 P appears rather simplistic. Can more of the data generated in the study be incorporated in it?

Response to Point 4: As recommended, we have incorporated more of the data, specifically the inclusion of Sox9 and OSX expression dependency on ERK from figures 1 and 2, and have done our best to account for the dose-dependency effects observed.

Reviewer 2 Report

Wnt-dependent activation of ERK mediates repression of chon-drocyte fate during calvarial development (Ibarra et al.,)

The authors showed in this paper that ERK activation leads to change of cranial mesenchymal cell fate via regulation of Sox9 expression. In detail, loss of ERK1/2 increased Sox9 positive cells in frontal bone primordia, resulting in less calvarial bone formation. Furthermore, the authors indicated that inhibition of b-catenin decreased pERK level and ectopic Sox9 expression was observed in this background. The author also examined several marker expressions in triple heterozygotes with ERK1/2 and b-catenin. Potentially this work is well designed and the authors actually showed key results for supporting the claims, but I feel that several points described below should be addressed before publication.

(major)

Fig. 1E-J: The authors should indicate the number of the samples they observed.

mRNA decreased, but IHC clearly showed that the number of Sox9(+)cell is increased. The author should mention about this discrepancy.

Fig. 2H-K’: Was OSX expression in E13.5 mice observed?

Fig. 3: How do the authors define the boundary between fbp and brain (shown in dotted lines). In Fig3E, Sox9 expression is seen between brain and fbp, whereas not in Fig. 3H, so the authors should rule out the possibility of the “shift” of expression domain. I think that precise position of a series of boundaries are extremely important to the authors’ claim.

Fig. 3H and J: Ectopic expression region of Sox9 seems different from that of Col2a1 (even though these samples are not same individuals).   

P11L13: The authors did not conduct the experiment for showing the involvement with non-canonical Wnt signaling, thus the sentence “considering~in vitro,” should be deleted.

Fig. 5P: The author should add Sox9 in this schematic diagram (b-catenin-|sox9->cartilage fate?).

Fig. 6: The authors claim functional link of two pathways for cartilage fate determination from this experiment, but it is hard to interpret the logics. This result only shows the dose dependency of these signaling for the fate determination. If the authors want to show the functional link, they should examine whether Wnt effect is suppressed by ERK overexpression, for example. These kinds of experiments may be important for evaluating the parallel contribution of these pathways for Sox9 expression/cartilage formation.

(minor)

P5L41-43: “relative to controls” is duplicated.

Fig. 2: figure labels I, I’, K and K’ are incorrect.

Author Response

(major)

Point 1: Fig. 1E-J: The authors should indicate the number of the samples they observed.

Response to Point 1: We have indicated the sample size within the figure legend in addition to the blanket statement listed within the methods.

Point 2: mRNA decreased, but IHC clearly showed that the number of Sox9(+)cell is increased. The author should mention about this discrepancy.

Response to Point 2: We have briefly addressed this within the text as well (Page 6 line 14). We believe that the discrepancies between Sox9 mRNA and protein levels may be due to posttranslational effects of ERK signaling on Sox9 protein stability or degradation (Williams et al 2020). Future studies will need to focus on the mechanism by which ERK regulates Sox9 protein expression.

Point 3: Fig. 2H-K’: Was OSX expression in E13.5 mice observed?

Response to Point 3: OSX expression in E13.5 mice is described in Figure1.

Point 4: Fig. 3: How do the authors define the boundary between fbp and brain (shown in dotted lines). In Fig3E, Sox9 expression is seen between brain and fbp, whereas not in Fig. 3H, so the authors should rule out the possibility of the “shift” of expression domain. I think that precise position of a series of boundaries are extremely important to the authors’ claim.

Response to Point 4: Considering the spatial restriction of ectopic Sox9 in our mouse models is limited to a distinct subset of cells, it is very important that the boundaries which delineate the frontal bone primordia (fbp) are consistent. We have also considered this point and we rely on two different reference points in the sections to create this boundary consistently. The boundary between the fbp and brain is defined based on both the expression domain of OSX shown in adjacent sections, as well as the morphological boundary between the condensed frontal bone mesenchyme as discerned by DAPI staining. Additionally, we rely on the meningeal layers to distinguish between the boundary of the brain and the fbp. We have revised our methods to address this point on page 4.

Point 5: Fig. 3H and J: Ectopic expression region of Sox9 seems different from that of Col2a1 (even though these samples are not same individuals).   

Response to Point 5: We observe the ectopic expression within the basal-medial region of the fbp similar to where we observe the ectopic expression. We have revised this figure to highlight the similarity. We believe that the expression domain are consistent with where we see ectopic Sox9 protein expression in the comparative section of the same animal. 

Point 6: P11L13: The authors did not conduct the experiment for showing the involvement with non-canonical Wnt signaling, thus the sentence “considering~in vitro,” should be deleted.

Response to Point 6: This phrase has been deleted as per reviewer’s suggestion.

Point 7: Fig. 5P: The author should add Sox9 in this schematic diagram (b-catenin-|sox9->cartilage fate?).

Response to Point 7: We have adjusted the schematic to address this concern and included sox9 as well as osx to improve the schematic diagram.

Point 8: Fig. 6: The authors claim functional link of two pathways for cartilage fate determination from this experiment, but it is hard to interpret the logics. This result only shows the dose dependency of these signaling for the fate determination. If the authors want to show the functional link, they should examine whether Wnt effect is suppressed by ERK overexpression, for example. These kinds of experiments may be important for evaluating the parallel contribution of these pathways for Sox9 expression/cartilage formation.

Response to Point 8: We thank the reviewer for this suggestion. We agree and have revised the manuscript to claim that there is dose-dependency and have removed any narrative about the functional link. While the proposed experiments would be undoubtedly insightful, they are outside the scope of the current study.

(minor)

Point 9: P5L41-43: “relative to controls” is duplicated.

Point 10: Fig. 2: figure labels I, I’, K and K’ are incorrect.

Response to Points 9/10: Both minor concerns have been addressed within the text and document.

Round 2

Reviewer 2 Report

There are no further concerns to address.